# Evaluation of Cas13d as a tool for genetic interaction mapping

Ghanem El Kassem, Jasmine Hillmer & Michael Boettcher ✉

Mapping genetic interactions (GIs) is crucial for understanding genetic network complexity. In this study, we investigate the utility of Cas13d, a CRISPR system targeting RNA, for GI mapping and compare it to Cas9 and Cas12a, two DNA nucleases commonly used for GI mapping. We find that Cas13d induces faster target gene perturbation and generates more uniform cell populations with double perturbations than Cas9 or Cas12a. We then encounter Cas13d gRNA-gRNA interference when concatenating gRNAs targeting different genes into one gRNA array, which we overcome by a dual promoter gRNA expression strategy. Moreover, by concatenating three gRNAs targeting the same gene into one array, we are able to maximize the Cas13d-mediated knockdown effects. Combining these strategies enhances proliferation phenotypes while reducing library size and facilitates reproducible quantification of GIs in oncogenic signaling pathways. Our study highlights the potential of Cas13d for GI mapping, promising advancements in understanding therapeutically relevant drug response pathways.

Understanding the intricate network of genetic interactions is fundamental to unraveling the complexity of cellular systems, deciphering disease mechanisms and identifying novel therapeutic targets. To this end, genetic interaction (GI) mapping has proven to be a powerful approach to investigate the functional relationships between genes[1,2]. GI mapping involves the pairwise perturbation of genes in the same cell, followed by the quantification of the resulting loss-of-function phenotypes to elucidate how one gene modulates the phenotype of the other. Initially, GI mapping was limited to model organisms, such as yeast[3,4] as it was not technically possible to perturb genes in human cells. This changed, with the advent of RNA interference (RNAi), allowing GI mapping to be performed in human cells, albeit with significant limitations such as off-target effects[5]. Over the last decade, more precise CRISPR-based approaches have almost completely replaced RNAi as a tool for GI mapping. Especially Cas9 nuclease of *Streptococcus pyogenes*[6–8] and to a lesser extent Cas12a (Cpf1) nuclease of *Acidaminococcus sp.*[9,10], combinations of both[11] as well as combinations of other orthologous CRISPR/Cas9 systems[12,13] have been used for mapping GIs in human cells.

In recent years, the development of RNA-targeting CRISPR effector proteins, such as Cas13, has expanded the toolbox for genetic perturbation experiments[14]. In particular, Cas13d from *Ruminococcus*

*flavefaciens*, a type VI-D CRISPR-Cas system, offers a versatile platform for programmable transcriptional interference, enabling precise and efficient manipulation of gene expression at the RNA level[15]. Several studies have since demonstrated the utility of Cas13d for genetic screens targeting coding[16] as well as non-coding transcripts[17], combinatorial Perturb-seq screens[18] and multiplexed transcriptomic regulation in primary human T cells[19]. However, no study to date has explored the utility of Cas13d for quantitative GI mapping.

To be applicable for GI mapping, a CRISPR system must be able to specifically and homogeneously perturb two genes within the same cell to generate a uniform double-perturbed cell population. The DNA-targeting CRISPR enzymes Cas9 and Cas12a introduce genetic perturbations into target sites by the introduction of a DNA double strand break which subsequently is repaired by endogenous cellular DNA-damage repair mechanisms, such as classical nonhomologous end joining (cNHEJ), microhomology-mediated end joining (MMEJ) and single-strand annealing (SSA)[20]. These repair processes are error-prone and therefore often lead to a large variety of INDEL mutations at the target site, resulting in a genetically diverse cell population with in-frame and frameshift mutations[21]. The RNA-targeting Cas13d system, on the other hand, does not depend on cellular DNA repair pathways, as it acts on the RNA transcript and should therefore cause uniform

Universitätsmedizin Halle, Martin Luther University Halle-Wittenberg, Halle (Saale), 06120 Halle, Germany. ✉ e-mail: michael.boettcher@medizin.uni-halle.de

reduction in target transcript levels throughout the entire cell population.

Another prerequisite for a CRISPR system to be applicable for GI mapping is the absence of sequence-specific interference between guide RNAs (gRNAs) targeting two different genes within the same cell. This is important because if the activity of one gRNA changes depending on the sequence of the second gRNA, quantification of GI scores, defined as the deviation of the measured phenotype of the double perturbation from the expected phenotype calculated from the measured individual perturbation phenotype of each gene, becomes impossible. In this context, it is important to point out that both, Cas12a[22] and Cas13d[15], in contrast to Cas9, can process their own gRNA arrays, which means that with these systems it is possible to express arrays of multiple concatenated gRNAs targeting different genes from the same promoter. Previous studies have demonstrated the applicability of Cas12a gRNA arrays for GI mapping[9,10]. While studies have used Cas13d gRNA arrays for multiplexed gene perturbation[18,19] an assessment of the suitability of Cas13d gRNA arrays for GI mapping is still lacking.

Here, we set out to explore the utility of Cas13d for GI mapping. We demonstrate that Cas13d induces genetic perturbations faster and more homogeneously than Cas9 or Cas12a respectively, thereby generating highly uniform populations of cells with double perturbations. Furthermore, our results show that the concatenation of Cas13d gRNAs in an array can lead to a modulation of the activity of one gRNA depending on the sequence of the second gRNA. This shows that concatenating Cas13d gRNAs targeting different genes in one array is not feasible for GI mapping. To address this issue, we show that expressing individual gRNAs from distinct promoters eliminates the sequence-specific interference. Moreover, we find that concatenating three Cas13d gRNAs targeting the same gene in one array, amplifies the efficacy of target gene knockdown and enhances proliferation phenotypes, while minimizing library size. Finally, we carry out a series of GI screens using either single gRNAs or single-gene arrays expressed from two promoters to systematically map GIs between six genes that modulate the response of the chronic myeloid leukemia cell line K562 to the tyrosine kinase inhibitor imatinib. We observe overall larger effect sizes in GI scores with single-gene arrays compared to single gRNAs. We find that GI scores were not only reproducible within each approach, but also correlated well between the single gRNA and single array screens. In summary, we successfully establish and benchmark two Cas13d approaches for the quantification of interactions between genes in therapeutically relevant oncogenic signaling pathways.

## Results

### Single gene perturbation properties of Cas9, Cas12a and Cas13d

In contrast to the DNA-targeting nucleases Cas9 and Cas12a, the ribonuclease Cas13d targets RNA for degradation (Fig. 1A). Consequently, the mechanisms by which these nucleases perturb their targets are fundamentally different, with Cas9 and Cas12a relying on the introduction of frameshift mutations through endogenous DNA double-strand break repair pathways, while Cas13d degrades its RNA targets autonomously. To analyze how these differences affect the performance of single and double perturbations, the perturbation kinetics between the three systems were compared. For that purpose, the components of the three CRISPR systems were introduced into the chronic myeloid leukemia cell line K562 via lentiviral transduction at a low multiplicity of infection (MOI < 0.3).

The example histograms in Fig. 1B illustrate how Cas13d reduces CD46 protein levels faster and more uniformly than the DNA-targeting nucleases Cas9 and Cas12a. These differences result from the different mechanisms of action of DNA- versus RNA-targeting CRISPR systems. The near-random nature of the DNA double-strand break repair outcomes responsible for the Cas9 and Cas12a mediated knockout results

in bimodal populations consisting of cells with null mutations and wild-type levels of CD46. Cas13d on the other hand generates a homogenous cell population with reduced CD46 levels, within three days post gRNA transduction, due to its RNA-degrading mechanism of action (Fig. 1B).

Next, cell surface markers CD46, CD47, CD63, and CD71 were targeted with Cas9, Cas12a, and Cas13d followed by flow cytometric quantification of target protein levels over time (Fig. 1C). Three days after transduction, little to no reduction in target protein levels was observed for either of the two DNA-targeting nucleases, while Cas13d showed an almost complete knockdown. Cas9 and Cas12a nucleases reached maximum target protein reduction at day 5 or day 7 post-transduction, depending on the target. Taken together, these results show that Cas13d can reduce target protein levels faster than Cas9 or Cas12a.

For the essential gene *CD71* (K562 DepMap Chronos score: -0.975), Cas9 and Cas12a mediated knockout showed a decrease of edited cell populations by day 7 or day 10, respectively. Cas13d on the other hand produced a stable knockdown between 79% and 87% during the entire course of the experiment. These observations suggest that complete deletion of CD71 by Cas9 and Cas12a impairs K562 cell survival, whereas a greater than 80% reduction in CD71 levels does not. Consequently, "fine-tuned" Cas13d knockdown could be useful to investigate the function of genes whose complete deletion is cytotoxic.

### Double gene perturbation properties of Cas9, Cas12a and Cas13d

The described differences in the mode of target gene perturbation between the three CRISPR systems are of particular importance when more than one gene is perturbed per cell, as is required for GI mapping. Figure 1D shows the distribution of cells after single or double gene perturbation from all three tested CRISPR systems. In case of Cas9 and Cas12a double perturbation, a mix of unperturbed, single-perturbed, and double-perturbed cell populations remained, while in case of Cas13d the whole cell population uniformly shifted towards double negative (Fig. 1D). Double-perturbed cell populations were 51.2% ±5 for Cas9, 81.6% ±0.7 for Cas12a and 95.4% ±0.9 for Cas13d (Fig. 1E). Taken together, these results demonstrate that Cas13d can not only produce more rapid gene perturbations than both DNA-targeting CRISPR systems, but also generates the most uniform, double-perturbed cell populations, making it a promising tool for GI mapping.

### Concatenated Cas13d gRNAs show sequence-dependent interference

Cas13d has the ability to process a concatenation of multiple gRNAs, called gRNA array, into functional single gRNAs[15]. Here, we wanted to assess the utility of Cas13d gRNA arrays for GI mapping, which requires the quantification of proliferation values (tau) for the calculation of GI scores. The GI score between two genes is calculated as the deviation of the measured phenotype of the perturbation of both genes in the same cell, from the expected phenotype calculated from the measured perturbation phenotype of each gene individually. Therefore, two gRNAs must act identically on their target RNA, regardless of the sequence of the second gRNA that is expressed in the same array.

In order to determine whether this is true for Cas13d arrays, gRNAs were designed against six genes whose involvement in the imatinib response of the chronic myeloid leukemia cell line K562 we had previously established[12]. K562 cells carry a translocation between chromosomes 9 and 22, which creates the *BCR::ABL* fusion oncogene, a constitutively active tyrosine kinase that causes the cells to divide in an uncontrolled fashion[23]. Application of BCR::ABL inhibitors, such as imatinib, have revolutionized the treatment of chronic myeloid

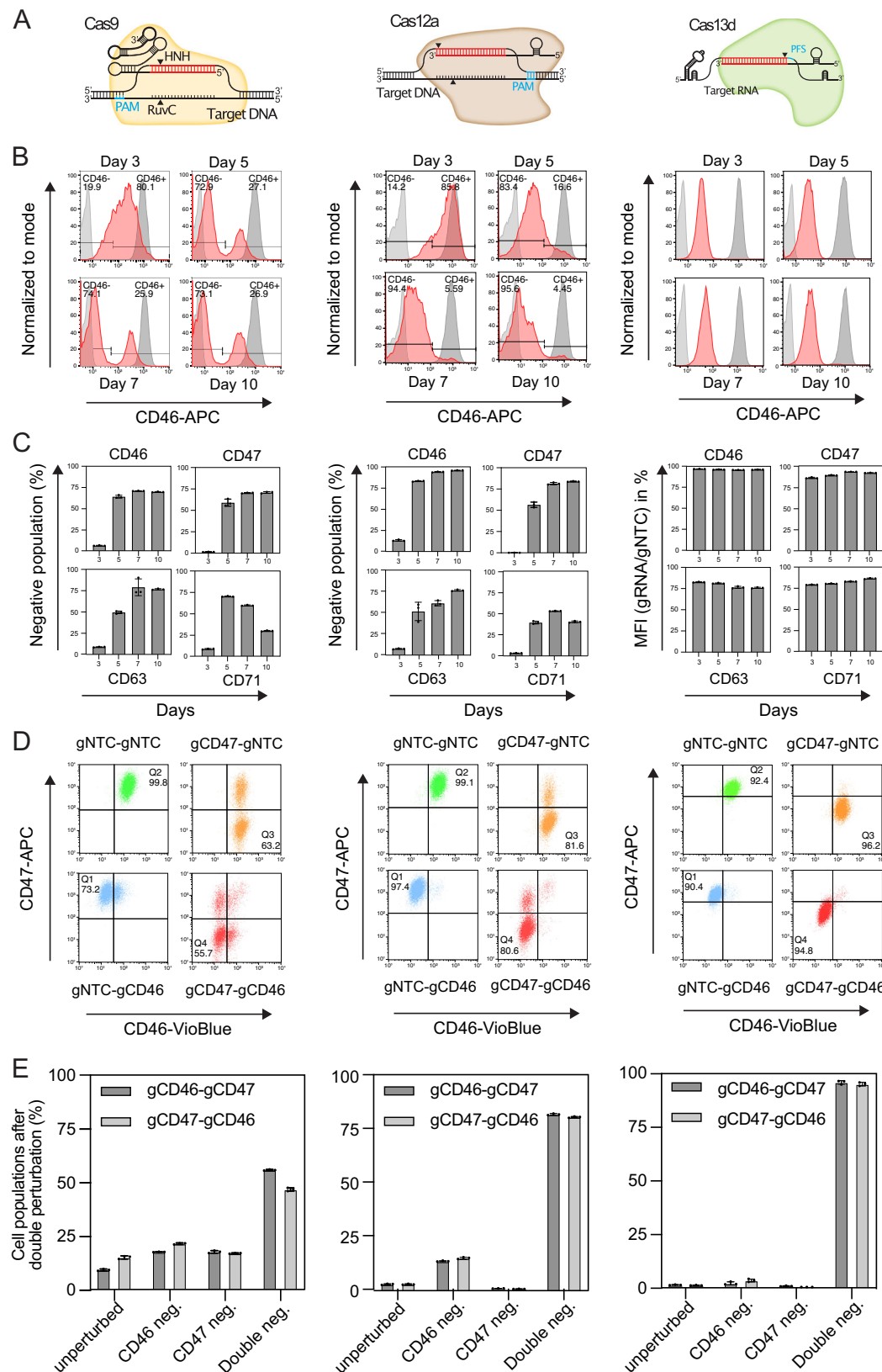

leukemia[24]. Three genes whose perturbation sensitized K562 cells to imatinib treatment, namely *(BCR)-ABL1*, the direct target of imatinib, *GAB2* and *SOS1* as well as three genes whose perturbation caused the cells to become less sensitive to imatinib, namely *PTPN1*, *NF1* and *SPRED2* were selected. 27 gRNAs against each target gene were designed, using Cas13d gRNA design rules previously described

(Supplementary Data 1)[16]. The dual gRNA array library was used for pooled CRISPR screens as described in Fig. 2A. The library was cloned in a way that two concatenated gRNAs were expressed from a single mouse U6 (mU6) promoter (Fig. 2B). Each dual gRNA array either targeted a gene in position 1, in combination with one of 30 different non-target control gRNAs (gNTC) in position 2 (U6-gRNA-gNTC), or

**Fig. 1 | Comparison of single gene and double gene perturbation properties of Cas9, Cas12a, and Cas13d. A** Schematic of the different CRISPR systems, targeting DNA (Cas9 and Cas12a) or RNA (Cas13d). **B** Histograms of cells perturbed with Cas9, Cas12a, and Cas13d targeting CD46 with protein levels measured via flow cytometry at four different time points post-transduction of the respective CRISPR system. Light gray: Unstained untransduced cells. Red: Perturbed cells. Dark gray: Stained non-target control cells. **C** Gene perturbation kinetics of Cas9, Cas12a, and Cas13d targeting CD46, CD47, CD63, and CD71 over 10 days. Values represent the mean of

biological replicates; error bars, SD ($n = 3$). **D** Distribution of cell populations expressing combinations of CD46, CD47 and NTC gRNAs as indicated. Cell surface marker levels were quantified via flow cytometry at 10 days post-transduction. **E** Quantification of unperturbed, single-perturbed, and double-perturbed subpopulations from cells expressing gRNAs against CD46 and CD47 at 10 days post-transduction. Values represent the mean of biological replicates; error bars, SD ($n = 3$). Source data are provided as a Source Data file.

vice versa (U6-gNTC-gRNA). Correlation between tau values from imatinib treated technical screen replicates was found to be high ($r = 0.79$), confirming the technical reproducibility of screen results (Fig. 2C, Supplementary Fig. 1A–E). Interestingly, comparison between tau values from the same gRNAs in both array positions, namely gRNA-gNTC (position 1) and gNTC-gRNA (position 2), revealed heterogeneous correlation patterns between both positions, suggesting position-dependent performance differences of the same gRNA (Fig. 2D, Supplementary Fig. 1F, Supplementary Data 2). To further explore this issue, the three best performing gRNAs against *PTPN1* and *SOS1* were selected and ranked, based on their performance in position 1 of the gRNA array, combined with 30 different gNTCs (Fig. 2E, top panel). Large variations of tau values were observed from the same gRNAs, depending on which gNTC sequence they were concatenated with in position 2. While for example the concatenation with all six selected gRNAs with gNTC-28 produced strong positive (gPTPN1) or negative (gSOS1) tau values, the same gRNAs concatenated with gNTC-21 produced only minimal enrichment or depletion of cells. In the opposite orientation (gNTC-gRNA) this trend was not observed (Fig. 2E, bottom panel). To obtain a more systematic overview, tau values from gene-targeting gRNAs, expressed either in position 1 (Supplementary Fig. 2B) or position 2 (Supplementary Fig. 2C) in combination with the 30 different gNTCs in the respective other position, were determined. Similar to the results shown in Fig. 2E, the effect sizes of tau values from specific gRNAs against a target gene in position 1 varied, in a gNTC-sequence specific manner, while no such effect was observed when the gene targeting gRNA was expressed in position 2. The strong positive and negative correlations between arrays with different gene-targeting gRNAs in position 1 further demonstrates the systematic gRNA-gRNA interference between gene-targeting gRNAs in position 1 (Supplementary Fig. 2D). The absence of systematic correlation between gene targeting gRNAs expressed in position 2, suggests that gRNA-gRNA interference happens only when the gene-targeting gRNA is expressed in position 1, but not 2, probably due to the way *Rfx*Cas13d processes gRNA arrays. Taken together, these results show that concatenation of Cas13d gRNAs against different target genes is not a viable option for quantitative GI mapping approaches due to sequence-dependent interference between gRNAs of the same array.

### Expression of gRNA from separate promoters prevents gRNA-gRNA interference

To overcome the gRNA-gRNA interference observed with concatenated gRNAs, a modified gRNA expression strategy was implemented, in which each gRNA was expressed from a separate promoter. To prevent recombination during library cloning, a mouse U6 promoter (mU6) was used to drive the expression of gRNA-1 and a human U6 promoter (hU6) for gRNA-2 (Fig. 2F). Using this setup, a pooled library was cloned, targeting the same six genes as before, by 27 gRNAs per gene either expressed from the mU6 promoter (position 1) or the hU6 promoter (position 2) (Supplementary Data 1). This library was used for a pooled screen under the same conditions described in Fig. 2A. Correlation between tau values of imatinib treated technical screen replicates was found to be high ($r = 0.91$), indicating the reproducibility of the screen data (Fig. 2G, Supplementary Fig. 3A–E).

Comparing the correlation of gene targeting gRNAs expressed in both positions, revealed similarly high correlation of 0.92 (Fig. 2H, Supplementary Fig. 3F, Supplementary Data 2) indicating that the gene targeting gRNA performance was independent of the position it was expressed from. Just like in the concatenated setup (U6-g1-g2) active gRNAs against all six target genes were identified. However, in the dual promoter setup (U6-g1-U6-g2), the observed phenotypes from active gRNAs remained consistent in both positions, regardless of the sequence of the co-expressed gNTC (Supplementary Fig. 2F, G). Moreover, no gNTC-sequence specific performance differences were observed between different gene-targeting gRNAs (Supplementary Fig. 2H), suggesting that the expression of two gRNAs from separate promoters can overcome the gRNA-gRNA interference observed from concatenated gRNAs above. These findings suggest the general usability of Cas13d for GI mapping.

### Single-gene arrays generate stronger knockdown and proliferation phenotypes

While gRNA-gRNA interference precludes the use of Cas13d arrays against two different target genes for GI mapping, concatenation of multiple gRNAs against the same target gene could still be a viable option to achieve stronger target gene knockdown. To test this hypothesis, three gRNAs against the same target gene were concatenated into one array, termed "single-gene array" (Fig. 2I). The knockdown effects of single-gene arrays were then compared to the effects of each of the three single gRNAs individually, by targeting the same cell surface markers CD46, CD47, CD71 and CD63 (Supplementary Data 3). Single-gene arrays consistently showed stronger knockdown effects than any of the three single gRNAs alone with residual target protein expression as low as 0.7% for CD46, 4.9% for CD47, 10% for CD71, and 17% for CD63 (Fig. 2J). To more systematically test the utility of single-gene arrays for GI mapping, the same 27 gRNA sequences used to generate the dual promoter single gRNA library above, were concatenated into 9 single-gene arrays (Fig. 2K, Supplementary Data 1). This led to a 9-fold reduction in library size from 36,864 elements to only 4,096 elements. These results demonstrate the utility of the Cas13d single-gene arrays in maximizing perturbation phenotypes while minimizing library size.

### Cas13d perturbation shows no non-specific proliferation phenotypes

Recent research has shown that under certain conditions, different Cas13 ribonucleases, including RfxCas13d used in this study, can induce non-specific RNA degradation in eukaryotic cells, potentially leading to proliferation defects in certain cell types[25]. To validate growth phenotypes resulting from Cas13d perturbations, Cas9 counter screens were conducted as described above (Fig. 2A). For this purpose, a Cas9 sgRNA library that targeted the same six genes as both Cas13d libraries, with four sgRNAs per gene was designed and cloned. Proliferation phenotypes determined from untreated cells and cells after 19 days of imatinib treatment showed high levels of correlation between the Cas9, Cas13d single gRNAs and single-gene arrays approaches (Supplementary Fig. 4). Moreover, comparison with Chronos scores from K562 Depmap data[26] revealed high levels of correlation with results from the

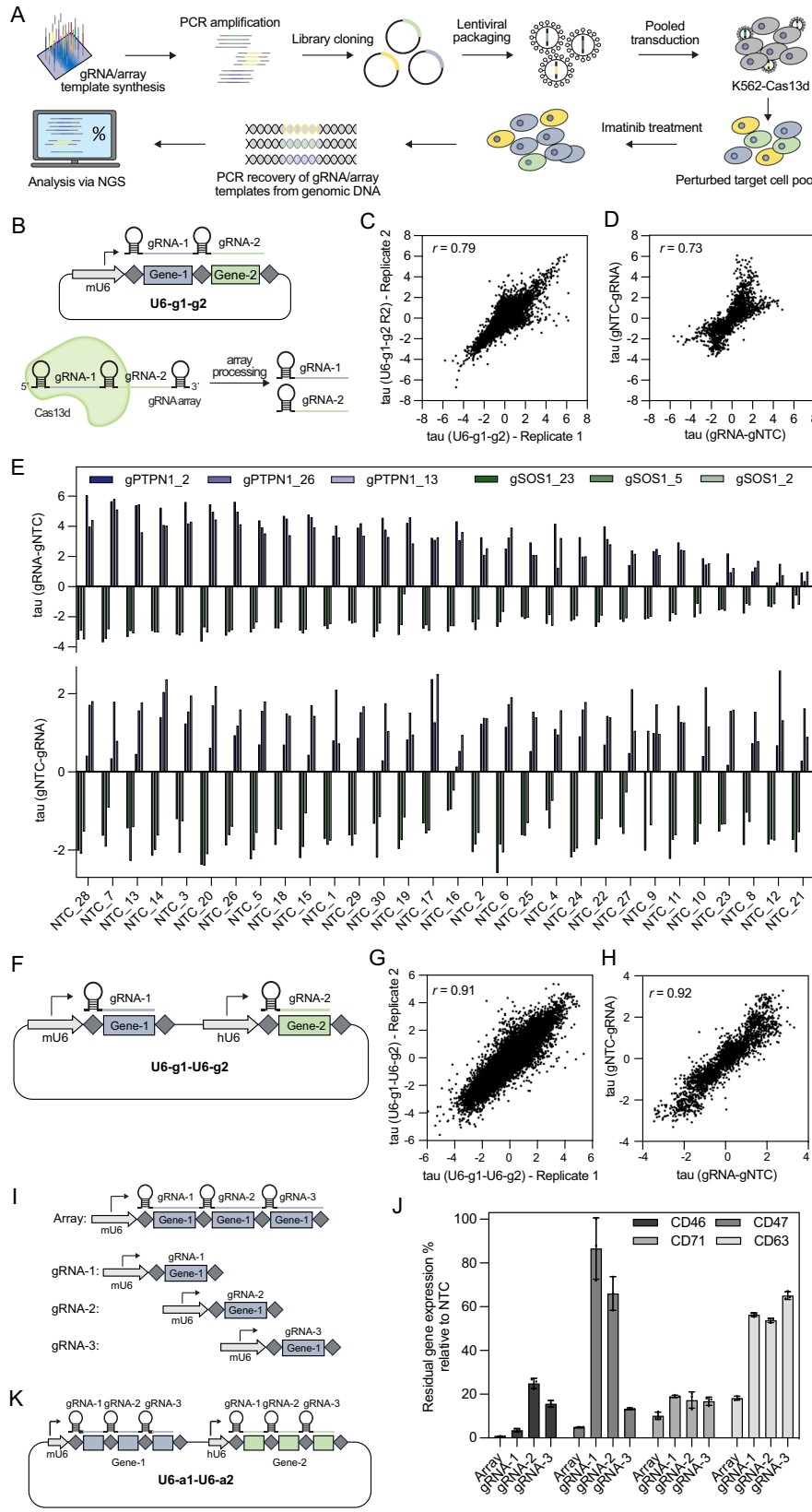

Cas9 and both Cas13d screens under untreated conditions. Most importantly, no instances of cell death were observed following Cas13d knockdown of a target gene that could not be confirmed by Cas9-mediated knockout and DepMap data. In conclusion, this suggests the absence of cytotoxicity attributable to potential collateral activity in this study.

## Cas13d allows highly reproducible quantification of GIs in therapeutically-relevant signaling pathways

Finally, pooled screens were conducted as described in Fig. 2A, using the single gRNA library (U6-g1-U6-g2) and the single-gene array library (U6-a1-U6-a2) with and without imatinib treatment. The tau values determined from all four screens were used to compute GI scores

**Fig. 2 | Concatenated Cas13d gRNAs show sequence-dependent interference which can be overcome by gRNA expression from separate promoters.**
**A** Overview of the pooled Cas13d screening pipeline. **B** Schematic of the dual gRNA concatenation strategy U6-g1-g2. mU6 = mouse U6 promoter, diamonds = direct repeat DR36, squares = target-specific spacer sequence. Cas13d can process concatenated gRNA arrays into single gRNAs. **C** Correlation between tau values from two technical screen replicates, following 19 days of imatinib treatment using the U6-g1-g2 strategy. Pearson correlation was used to determine the *r* value.
**D** Correlation between tau values from gRNA-gNTC and gNTC-gRNA combinations following 19 days of imatinib treatment using the U6-g1-g2 strategy. Pearson correlation was used to determine the *r* value. **E** Bar chart of the tau values from the three best-performing gRNA targeting PTPN1 and SOS1 in combination with 30 gNTCs in both orientations: gRNA-gNTC (top) and gNTC-gRNA (bottom).
**F** Schematic of the dual promoter gRNA expression strategy mU6-g1-hU6-g2. mU6 = mouse U6 promoter, hU6 = human U6 promoter, diamonds = direct repeat DR36,

squares = target-specific spacer sequence. **G** Correlation between tau values from two independent screen replicates, following 19 days of imatinib treatment using the U6-g1-U6-g2 strategy. Pearson correlation was used to determine the *r* value.
**H** Correlation between tau values from gRNA-gNTC and gNTC-gRNA combinations following 19 days of imatinib treatment using the U6-g1-U6-g2 strategy. Pearson correlation was used to determine the *r* value. **I** Schematic of the single-gene array.
**J** Flow cytometric quantification of CD46, CD47, CD71, and CD63 residual protein levels in cells expressing either a single-gene array or one of the three single gRNAs that were concatenated in the single-gene array. Values represent the mean of biological replicates; error bars, SD (*n* = 3). **K** Schematic of the dual promoter single-gene array expression strategy used for subsequent U6-a1-U6-a2 libraries. mU6 = mouse U6 promoter, hU6 = human U6 promoter, diamonds = direct repeat DR36, squares = target-specific spacer sequence. Source data are provided as a Source Data file.

(Supplementary Data 2). In line with stronger knockdown effects observed from single-gene arrays when compared to single gRNAs (Fig. 2J), effect sizes of tau values and GI scores from single-gene arrays were also larger (Fig. 3A). Correlation between tau values of imatinib treated technical screen replicates was found to be high (*r* = 0.94), indicating the reproducibility of the screen data (Supplementary Fig. 5A–F). Comparing the correlation of gene targeting gRNAs expressed in both positions, revealed similarly high correlation of 0.9 (Supplementary Fig. 5H, Supplementary Data 2) indicating that the single gene targeting array performance was independent of the position it was expressed from.

Particularly in the untreated condition, only two GIs (*NF1-SOS1* and *NF1-GAB2*) were detected in both orientations with the single gRNA strategy while three additional GIs were detected with the single-gene array approach, such as the well characterized buffering interaction between the ABL1 kinase and its antagonist, the phosphatase PTPN1[27,28]. In imatinib-treated conditions, the effect sizes of GI scores were generally larger compared to untreated conditions. Like in the untreated condition, single-gene arrays outperformed single gRNAs by producing larger effect sizes for tau values and GI scores. In all four screens, "single gene controls" were calculated as the interaction between each of the six candidate genes with all non-target control gRNAs. As expected, none of the six investigated genes showed a significant genetic interaction with non-target controls (Fig. 3B).

Correlation of GI scores was found to be high within either approach as well as between both approaches, showing that both, the single gRNA and the single-gene array approach reproducibly picked up the same types of GIs (Fig. 3C and Supplementary Fig. 6). Notably, a subtle albeit consistent negative correlation was detected between GI scores under treated and untreated conditions in both approaches. This implies that some GIs change their nature under imatinib exposure, like for example, the aforementioned buffering GI between *ABL1* and *PTPN1*, which changes to synergistic when the cells are exposed to imatinib. In contrast, the synergistic interaction between *ABL1* and *SOS1* becomes buffering under imatinib selection. Figure 3D shows a de-novo generated interaction network, consisting of all reproducible GIs calculated from the tau values that were generated by the single-gene array approach in both gene orientations (Gene1-Gene2 and Gene2-Gene1) under untreated and imatinib treated conditions.

## Discussion
The study presented here explores the utility of Cas13d, a type VI-D CRISPR system that targets RNA, for quantifying genetic interactions (GIs). Mapping GIs is crucial for understanding the functional complexity within genetic networks. Previous GI mapping relied primarily on the DNA-targeting nucleases Cas9 and Cas12a, which have certain limitations, such as slow and heterogeneous editing kinetics (Fig. 1B–E). This complicates studying the function of essential genes

for example, such as *CD71* (K562 DepMap Chronos score: -0.975). Cas13d achieved 79% reduction of CD71 protein levels within 3 days post-transduction, without signs of CD71-perturbation related cytotoxicity over 10 days, as it was observed after Cas9 as well as Cas12a mediated knockout of CD71 (Fig. 1C). Moreover, Cas13d has been shown to effectively target non-coding RNAs for degradation, such as lncRNAs, which cannot be targeted by DNA-nuclease-induced reading frameshift mutations, since the function of lncRNAs does not rely on a reading frame[15,17].

Further, we found that in comparison to the RNA-targeting Cas13d, DNA-targeting nucleases generate bimodal populations of full knockout (null mutation) cells and wildtype cells (Fig. 1B). This poses a challenge for uncovering GIs that are relevant for therapeutic intervention because typically, pharmacological inhibitors reduce rather than completely abolish the activity of their targets. Unlike the DNA-targeting nucleases, Cas13d generated uniform knockdown populations which mimic this reduction in target activity. Moreover, the uniform knockdown mediated by Cas13d led to a higher fraction of double perturbed cell populations in comparison to both DNA-targeting nucleases (Fig. 1E). This property becomes particularly important for higher-order genetic interaction studies involving the perturbation of more than two genes in the same cell, emphasizing the significance of Cas13d in such endeavors. In this context, it is also important to mention that Cas9-induced DNA double-strand breaks can cause cytotoxicity by triggering the p53 response[29,30], and that this effect strongly correlates with the number of target loci[31,32]. Consequently, the induction of DNA double-strand breaks at multiple genomic loci is a potential problem for higher-order GI screens with DNA-targeted nucleases.

While cytotoxicity due to DNA double-strand breaks is no concern for RNA-targeting CRISPR systems, collateral RNA cleavage by Cas13d from *Ruminococcus flavefaciens* (*Rfx*Cas13d) used in this study, has been reported to induce varying levels of cytotoxicity in eukaryotic cells. The effect seems to depend on cell type, target gene expression levels and the delivery method of the *Rfx*Cas13d system[33–35]. While collateral RNA cleavage was not observed in the original *Rfx*Cas13d publication by Konermann et al. [15], it represents a potential concern for the use of *Rfx*Cas13d that should be carefully monitored in future studies. An overwhelming majority of studies however, have utilized *Rfx*Cas13d without reporting any adverse effects from collateral activity[16–19,36–40]. Similarly, in this study, we observed no unexpected growth phenotypes that could be attributed to collateral RNA cleavage (Supplementary Fig. 4) nor did we detect degradation of CD47 after perturbation of CD46, or vice versa, as it would be expected from collateral RNA trans-cleavage (Fig. 1D). To mitigate the effects of collateral activity, newer high-fidelity variants of *Rfx*Cas13d with minimal collateral activity have been described[41], offering promising options for future experiments in sensitive contexts. Additionally, Cas13d orthologs, such as *Dj*Cas13d, have been shown to exhibit strong

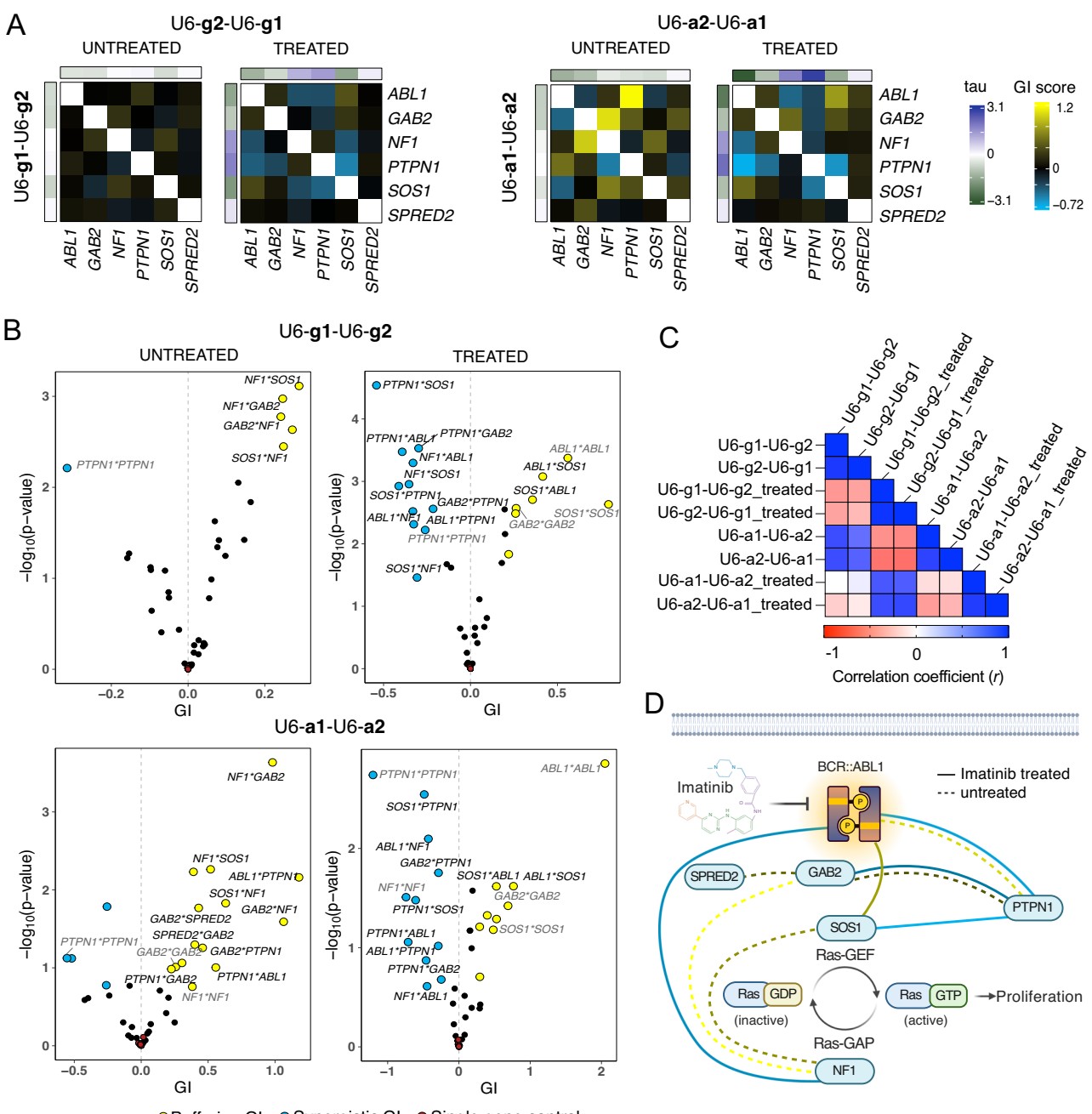

**Fig. 3 | Cas13d enables the highly reproducible identification of GIs in oncogenic signaling pathways. A** Genetic interaction maps showing GI scores from single gRNA (left panel) and single-gene array screens (right panel), without and with imatinib treatment. Tau values from single gene perturbations in both possible orientations (Gene1-gNTC and gNTC-Gene1) are displayed along the edges of each GI map. GI scores derived from gRNA/array orientation 1 are shown in the bottom left of each GI map, while GI scores derived from gRNA/array orientation 2 are shown in the top right. Positive GI scores indicate buffering interactions while negative GI scores indicate synergistic interactions. **B** Volcano plots showing GI scores and associated significance ($-\log_{10}$ ($p$-value)) for all possible gene-gene combinations from single gRNA (top) and single-gene array screens (bottom), with and without imatinib treatment. Blue and yellow data points indicate interactions that meet the threshold of GI $> < \pm 0.2$ and FDR $< 0.5$. Data points with black labels indicate GIs that were identified in both orientations (gRNA-gNTC and gNTC-gRNA). Data points labeled in gray indicate "same gene GIs" that passed the threshold, where gRNAs in both positions targeted the same gene. Red data points

indicate 'single gene controls' where GI scores were calculated between genes, with one gene replaced by non-target control gRNAs (see methods for details). Significance was calculated using limma moderated $t$ test followed by Benjamini–Hochberg multiple testing correction. **C** Pearson correlation ($r$) between GI scores determined from imatinib treated and untreated cells via the single gRNA and single-gene array approach, in both orientations. **D** Genetic interaction network of reproducible GIs in both gene orientations (Gene1-Gene2 and Gene2-Gene1) that were identified between all six investigated genes derived from the single-gene array strategy. Only interactions that were identified by GIs of $> < \pm 0.2$ and FDR $< 0.5$ in both orientations (gRNA-gNTC and gNTC-gRNA) are shown. Ras-GEF = Ras guanine nucleotide exchange factor. Ras-GAP = Ras-GTPase activating protein. Edges between genes are colored based on the average GI score of screen replicates and gRNA-gNTC orientations. Created in BioRender. Böttcher, M. (2025) https://BioRender.com/e57j701. Source data are provided as a Source Data file.

on-target activity with minimal cytotoxicity, even when targeting highly abundant transcripts[35].

Although the strategy of concatenating gRNAs against different target genes for GI mapping has been successfully applied with Cas12a[9,10], we detected sequence-specific interference between Cas13d gRNAs against target genes expressed in position 1 with non-target gRNAs expressed in position 2 of the same array (Fig. 2D, E, Supplementary Fig. 2B–D), rendering this approach ineffective for quantitative GI mapping. Further exploration of the underlying reasons may enable the future use of concatenated Cas13d gRNAs against different target genes for quantitative GI mapping. By implementing a two-promoter gRNA expression strategy, we were able to overcome the problem of gRNA-gRNA interference and make Cas13d applicable for quantitative GI mapping (Fig. 2F–H, Supplementary Fig. 2F–H). Furthermore, we show that concatenating three gRNAs against the same target gene not only generates stronger knockdown (Fig. 2J) and growth phenotypes (Fig. 3A) compared to single gRNAs, but also reduced the size of the combinatorial gRNA library by 9-fold, from 36,864 gRNA-gRNA combinations to only 4096 array combinations, while enhancing GI effect sizes at the same time (Fig. 3B). We further show that GI scores derived with both approaches were strongly concordant (Fig. 3C). The more compact size of the array library allows this approach to be readily adapted to study GIs between hundreds of genes in a single screen.

Last but not least, we benchmarked both approaches, the single gRNA (U6-g1-U6-g2) and the single-gene array approach (U6-a1-U6-a2) to study GIs between six genes involved in oncogenic signaling downstream of *BCR::ABL*. The determined GIs were highly reproducible both within and between both approaches (Fig. 3C). Interestingly, 3 out of the 4 synergistic GIs that could be detected under imatinib treatment were between *PTPN1* and *ABL1*, *SOS1* or *GAB2*, respectively (Fig. 3D), suggesting a negative upstream regulatory function of PTPN1 on these genes. The negative regulation of ABL1 by PTPN1 through direct dephosphorylation is well documented[27,28]. While no direct interaction between PTPN1 and the adaptor protein GAB2 or the Ras-GEF SOS1 has been described yet, the observed synergistic interactions could be explained by the well established function of both proteins as direct physical interaction partners of BCR::ABL, the target of PTPN1[42]. In addition, a synergistic interaction between *ABL1* and *NF1* was detected, which could be attributed to the opposing regulation of *RAS* activity by both genes. Consistent with its role as positive downstream effector of *ABL1*, a buffering GI was detected between *ABL1* and *SOS1* under imatinib selection, meaning that their simultaneous perturbation rendered the cells less sensitive than expected, possibly due to their involvement in the same signaling complex[43].

In conclusion, our study highlights the potential of Cas13d for GI mapping, enabling rapid knockdown and uniform population responses. Our strategy using dual promoters and concatenation of gRNAs against the same gene demonstrates robustness and efficiency in quantifying interactions between genes in oncogenic signaling pathways. These results show the promise of Cas13d in advancing our understanding of drug response pathways and identifying targets for therapeutic intervention.

## Methods
### Vector maps
The plasmids pXR001-mCD4, 783-Rx-hU6, 783-Rx-mU6, 783-Rx-Dual, pMB1, and AiO-Cas12a were used. Plasmids and their sequences are deposited at Addgene.

### K562 and Lenti-X 293T cell culture
K562 (ATCC, CCL-243) cells were cultured in complete RPMI supplemented with 10% fetal bovine serum (Sigma-Aldrich, 11875093) and 1% antibiotics (pen/strep). Lenti-X 293T cells (Takara, 632180) were cultured in complete DMEM (Thermo Fischer Scientific, 11995073) supplemented with 10% fetal bovine serum and 1% antibiotics (pen/strep).

### K562-Cas13d-mCD4 clonal line
pXR001 plasmid[15] was digested with NheI (NEB, R0131). The *mCD4* gene was amplified from the *S. aureus* Cas9 nuclease vector (Addgene #105998) using Phusion Flash High-Fidelity PCR Master Mix (ThermoFisher Scientific, F548L) according to the manufacturer's protocol. P2A fragment and Gibson Assembly compatible overhangs to the digested pXR001 plasmid site were added using the following primers: 5′-CGATGTTCCAGATTACGCTGGATCCGGCGCAACAAACTTCTCTCTG CTGAAACAAGCCGGAGATGTCGAAGAGAATCCTGGACCGCCGGACAT GTGCCGAGC-3′ and 5′-GCCCTCTCCACTGCCGCCCTGGCGCTGTTG GTGCCGG-3′. The amplified PCR fragment was purified from a 1% Agarose gel using NucleoSpin columns (Macherey-Nagel, 740609.250) and cloned into the digested pXR001 plasmid using Gibson Assembly[44].

Cas13d-mCD4 gene was introduced into the K562 cell line via lentiviral transduction. The cells were stained with an anti-mouse CD4 antibody (Miltenyi, 130-116-526) and single clones were sorted using a BD Melody Flow Cytometer 19 days after transduction. The single clones were expanded for 5 weeks. To test the functionality of the expanded clonal lines, we transduced the cells with a gRNA to knockdown CD46 (5′-CAGACAATTGTGTCGCTGCCATC-3′). The clonal lines were screened for the functionality of the Cas13d system after 10 days via flow cytometry analysis of >10,000 cells stained with CD46 antibody (Miltenyi, 130-104-509). The best performing clonal line out of 24 clonal lines was used for the CRISPR screens and further experiments.

### Cas9 gRNAs design and cloning for kinetics and dual knockout experiments
The web tool CRISPick[45,46] was used to select 4 *sp*Cas9 sgRNAs targeting *CD46*, *CD47*, *CD63*, and *CD71*. The sgRNAs were ordered as complimentary single-stranded oligos from Sigma-Aldrich using the following structure for the sense and the antisense strands respectively: TTGGNNNNNNNNNNNNNNNNNNNNNNN and AAACNNNNNNNNN NNNNNNNNNNNN. The complementary sequences were then joined by incubating 10 μM of each oligo in T4 ligation buffer in a 10 μL reaction in a thermocycler using the following program: 37 °C for 30 min, 95 °C for 5 min, and ramp down at 0.1 °C/s from 95 °C to 25 °C. In parallel, pMB1 vector was digested with AarI (Thermo Fischer Scientific, ER1582) and the linear plasmid was purified from a 1% agarose gel. The annealed oligos and the digested plasmid were then joined by T4 ligation. For that, the annealed oligos were diluted 1:200 with water and 1 μL was mixed with 1 μL 10x T4 ligation buffer and 5 U T4 DNA Ligase (Thermo Fisher Scientific, EL0011) in a 10 μL reaction. The mix was incubated at 16 °C overnight. The next day, 2 μL were used for transforming chemo-competent *DH5α*. Single colonies were then picked from the LB agar plates and the plasmids were isolated from liquid cultures using NucleoSpin columns (Macherey-Nagel, 740588.250).

To test the functionality of the cloned gRNAs, we transduced the HEK293ΔRAF1:ER[47] cells with the *CD46* gRNAs and K562 cells with the *CD47*, *CD63*, and *CD71* gRNAs. The cells were screened after 20 days for the knockout of *CD46* and 7 days for the knockout of *CD47*, *CD63*, and *CD71* via flow cytometry analysis of >10,000 cells stained with CD46-APC, CD47-APC, CD63-APC, or CD71-APC antibodies (Miltenyi, 130-104-558, 130-123-315, 130-118-151, 130-115-030). The best performing sgRNAs were used for further experiments (Supplementary Fig. 7 and Supplementary Data 3).

For the dual knockout of *CD46* and *CD47*, the sgRNAs in the first position were expressed from a mU6 promoter and the one in the second position from an H1 promoter. Modified Cas9 tracr sequences

WCR2 and VCR1L were selected to avoid recombination[48]. The pMB1 plasmid was digested with PaqCI and NheI (NEB, R0745S, R0131) and the linear plasmid was purified from a 1% agarose gel. Gene fragments with Gibson assembly compatible overhangs of the following structure gRNA1-WCR2-filler-VCR1L-gRNA2-H1promoter were ordered from Twist Biosciences. The gene fragments were PCR amplified using Phusion Flash High-Fidelity PCR Master Mix (ThermoFisher Scientific, F548L) according to the manufacturer's instructions. The amplified PCR fragment was purified from a 1% Agarose gel using NucleoSpin columns (Macherey-Nagel, 740609.250) and cloned into the digested pMB1 plasmid using Gibson Assembly[44].

## AiO-Cas12a plasmid cloning

The all-in-one (AiO) Cas12a plasmid is composed of enAsCas12a gene expressed from an EF1α promoter and a gRNA expression cassette from a mU6 promoter. To clone the plasmid, enAsCas12a gene was PCR amplified from the pCAG-enAsCas12a plasmid[49] using a nested PCR strategy. The PCR reactions were performed using the Phusion Flash High-Fidelity PCR Master Mix (ThermoFisher Scientific, F548L) according to the manufacturer's instructions using the following primers for the first PCR reaction: 5′-ACCGGTTCTAGAGCGATGC CTGCTGCTAAGAGAGTGAAACTGGATCCTGCTGCTAAGAGAGTGAAA CTGGATCCTGCTGCTAAGAGAGTGAAACTGGATACACAGTTCGAGGG CTTTACC-3′ and 5′-GGATCCGCTAGCGTTGCGCAGCTCCTGGA-3′. For the second PCR reaction, Gibson Assembly compatible overhangs were added to the PCR fragments of the first PCR reaction using the following primers: 5′-AACACAGGACCGGTTCTAGACTAGAGCGATGCC TGCT-3′ and 5′-AGAGAGAAGTTTGTTGCGCCGGATCCGCTAGCGT TGCG-3′. CROP-Seq Cas9 Plasmid was digested with XhoI and BamHI (NEB, R0146S, R0136S) and the linear plasmid was purified from a 1% agarose gel. The enAsCas12a gene was cloned into the digested CROP-Seq Cas9 Plasmid backbone using Gibson Assembly[44]. The resulting plasmid was then digested with SacII and SnaBI (NEB, R0157S, R0130S). Gene fragment containing WPRE-3LTR with Gibson Assembly compatible overhangs was ordered from Genscript and cloned in the digested plasmid. The resulting plasmid was then digested with XhoI and IlluminaPBS-Filler-mU6 gene fragment (Genscript) that was PCR amplified using the following primers: 5′-GATCCACTTTGGCGC CGGCCTCGAGCAG-3′ and 5′-CTTTCAAGACCTAGGGCCCCCCTCGAG CCCGGGCATGCTCTTCAACCTCAATAACTGGAGTTATATGGACCATT GTTCTAGCGCTGATCCGACG-3′. The PCR reaction was performed using the Phusion Flash High-Fidelity PCR Master Mix (ThermoFisher Scientific, F548L) according to the manufacturer's instructions. The fragment was then inserted in the digestion site with Gibson Assembly. The final plasmid contained the enAsCas12a gene linked to the puromycin resistance gene under the EF1α promoter and the gRNA expression cassette under the mU6 promoter in the reverse orientation.

## Cas12a gRNAs design and cloning for kinetics and dual knockout experiments

Cas12a gRNAs targeting CD63 and CD47 used by DeWeirdt et al.[9] to assess and optimize Cas12a performance for combinatorial screens were selected. Cas12a gRNAs targeting CD46 and CD47 were selected using the web tool CRISPick[45,46]. The gRNAs for single gene knockout were ordered as complimentary single-stranded oligos from Sigma-Aldrich using the following structure for the sense and the antisense strands respectively: AAAANNNNNNNNNNNNNNNNNNNNNNN and TTGGNNNNNNNNNNNNNNNNNNNNNNN. For dual gene knockout, the gRNAs were arranged as 2 gRNA arrays separated by a DR sequence and complimentary single-stranded oligos with 4 nucleotide T4 ligation overhangs were ordered as before from Sigma-Aldrich (Supplementary Data 3). The complementary sequences were then joined by incubating 10 μM of each oligo in T4 ligation buffer in a 10 μL reaction in a thermocycler using the following program: 37 °C

for 30 min, 95 °C for 5 min, and ramp down at 0.1 °C/s from 95 °C to 25 °C. The AiO-Cas12a plasmid was digested with BsmBI-v2 (NEB, R0739S) overnight followed by purification from a 1% agarose gel using NucleoSpin columns (Macherey-Nagel, 740609.250). The annealed oligos and the digested plasmid were then joined by T4 ligation. For that, the annealed oligos were diluted 1:200 with water and 1 μL was mixed with 1 μL 10x T4 ligation buffer and 5 U T4 DNA Ligase (Thermo Fisher Scientific, EL0011) in a 10 μL reaction. The mix was incubated at 16 °C overnight. The next day, 2 μL were used for transforming chemo-competent DH5α. Single colonies were then picked from the LB agar plates and the plasmids were isolated from liquid cultures using NucleoSpin columns (Macherey-Nagel, 740588.250).

## Cas13d single gRNAs and arrays cloning

Cas13d gRNAs targeting CD46, CD47, CD63, and CD71 were selected from cas13design.nygenome.org[16]. The single gRNAs were ordered as single-stranded oligos with Gibson Assembly compatible overhangs (Sigma Aldrich). The oligos had the following structure: ACTGG TCGGGGTTTGAAAC-(N)$_{23}$-CAAGTAAACCCCTACCAACTGGTCGGGGT TTGAAACTTTTTTTGAATTGGCCGCG. Arrays targeting the one gene were designed by concatenating 3 single gRNAs separated by Cas13d DR36, Gibson assembly overhangs were added as before, and the arrays were synthesized by Genscript as gene fragments. The oligos were PCR amplified using the Phusion Flash High-Fidelity PCR Master Mix (ThermoFisher Scientific, F548L) according to the manufacturer's instructions. gRNA-specific primers were used to avoid unspecific priming (Supplementary Data 3). The PCR-amplified DNA fragments were purified using NucleoSpin columns (Macherey-Nagel, 740609.250). The 783-Rx-hU6 plasmid was digested with BfuAI (NEB, R0701S) and purified from 1% agarose gel. The single gRNAs and arrays were cloned into the digested 783-Rx-hU6 Plasmid backbone using Gibson Assembly[44]. The gRNAs and arrays were expressed from a hU6 promoter.

## Cas13d gRNAs design and cloning for kinetics and dual knockdown experiments

The plasmids expressing CD46, CD47, CD63, and CD71 arrays were used for the Cas13d knockdown kinetics experiments (Supplementary Data 3). For dual gene knockdown, 783-Rx-Dual plasmid was cloned to express gRNAs from mU6 and hU6 promoters. To clone the plasmid, we removed the saCas9 and spCas9 tracrRNA sequences from the sgLenti-orthogonal vector (Addgene #105997). For this, the plasmid was digested with NheI and BfuAI overnight (NEB, R0131, R0701S) and the digested plasmid was purified from a 1% agarose gel using NucleoSpin columns (Macherey-Nagel, 740609.250). A gene fragment without the saCas9 tracrRNA was synthesized by Genscript and cloned in the plasmid cut position by Gibson Assembly. The resulting plasmid was then digested with SphI and PaqCI (NEB, R0182S, R0745S) and purified as before. A gene fragment without the spCas9 tracrRNA was synthesized by Genscript and cloned in the plasmid cut position by Gibson Assembly (Supplementary Data 3). To insert Cas13d DR36 sequences and gRNA insertion sites after the mU6 and hU6 promoters, the plasmid was digested with BfuAI (NEB, R0701S). Single-stranded DNA oligos containing the gRNA insertion sites were synthesized as oPools (IDT) (Supplementary Data 3). The DNA oligos were amplified using the Phusion Flash High-Fidelity PCR Master Mix (ThermoFisher Scientific, F548L) according to the manufacturer's instructions using the following primers: 5′-GCCGTCTAATGTTCAGCTAGTATGCAC AGTTGATCCGTCTC-3′ and 5′-GTGTGACGTATGATCAGATCTATGCTA CAGTGAACCGTCTC-3′. After amplification, the DNA fragments were purified using NucleoSpin columns (Macherey-Nagel, 740609.250) and digested with BsmBI-v2 (NEB, R0739S) overnight. The digested fragments and plasmid were annealed together using T4 Ligation (Thermo Fisher Scientific, EL0011).

Cas13d arrays targeting CD46 and CD47 were selected for dual knockdown of both genes. The arrays were synthesized as oligo pools (IDT). The arrays had the following structure: 5′-AGTATGCA-CAGTTGATCCGTCTCAAAAC-$(N)_{23}$-DR36-$(N)_{23}$-DR36-$(N)_{23}$-CAAGAGA-GACGGTTCACTGTAGCA-3′. The DNA oligos were amplified using the Phusion Flash High-Fidelity PCR Master Mix (ThermoFisher Scientific, F548L) according to the manufacturer's instructions using the following primers: 5′-GCCGTCTAATGTTCAGCTAGTATGCACAGTTGA TCCGTCTC-3′ and 5′-GTGTGACGTATGATCAGATCTATGCTACAGTGA ACCGTCTC-3′. After amplification, the DNA fragments were amplified using NucleoSpin columns (Macherey-Nagel, 740609.250) and digested with BsmBI-v2 (NEB, R0739S) overnight. Oligos were then cloned into the plasmid in both positions for combinatorial knockdown of CD46 and CD47 in a 2-step digestion and T4 ligation cycles. In short, the plasmid was first digested with BfuAI (NEB, R0701S) and the first position gRNA was cloned with T4 ligation. Then the plasmid was digested again with AarI (Thermo Fisher Scientific, ER1582) and the second position gRNA was cloned.

## Lentivirus production

Lenti-X 293T cells (Takara, 632180) were seeded at 65,000 cells per cm$^2$ in 25 mL media (DMEM, 10% FBS, 1% P/S) in a 15 cm dish and incubated overnight at 37 °C, 5% CO2. On the next day, 15 µg sgRNA library plasmid, 6 µg psPAX2 (Addgene #12260), 6 µg pMD2.G (Addgene #12259) and 108 µL Turbofect (Thermo Fisher Scientific, R0532) were mixed into 5.4 mL serum-free DMEM (Thermo Fischer Scientific, 11995073), vortexed briefly, incubated for 20 min at RT, and added to the cells. At 48 and 72 h post-transfection, the supernatant was harvested, passed through 0.45 um filters (Millipore) and 50x concentrated using PEG8000 lentivirus concentrator solution. The concentrator solution was composed of 80 g PEG-8000 (Sigma-Aldrich, 81268-1KG), 1.4 g NaCl (Carl Roth, 3957.1) in 80 mL MillQ water and 20 mL of 10x PBS (Bio-Rad, 1610780) (pH7.4) in 200 mL water. Concentrated lentivirus aliquots were stored at −20 °C.

## Knockout and knockdown efficiency assessment with flow cytometry

K562-CasRx cells were seeded in 12 well plates at 50,000 cells /mL. The cells were transfected with lentivirally packaged gRNA constructs at a low multiplicity of infection (MOI) = 0.2 and incubated at 37 °C, 5% CO2 for 48 h. After incubation, cells were selected with 2 µg/mL puromycin for 3 days. After 10-12 days of infection, the cells were stained with the respective antibodies against the target genes, and the knockout/knockdown efficiency was determined via flow cytometry analysis of >10,000 on a BD LSRFortessa II flow cytometer.

## Pooled screens libraries design

*ABL1*, *GAB2*, *SOS1*, *NF1*, *PTPN1*, and *SPRED2* genes were selected as target genes based on the combinatorial screen results presented in Boettcher et al. [12]. For the Cas13d combinatorial perturbation screen U6-g1-g2 and the U6-g1-U6-g2, 27 gRNAs were selected from the cas13design.nygenome.org algorithm. The gRNAs were selected based on 3 different criteria. The first and third groups of 9 gRNAs per gene were selected based on the cas13design algorithm predicted guide score. For the second group of gRNAs, we selected the gRNA sequences that target the highest number of the most expressed mRNA transcripts of a gene while also taking into account the guide score. For that, the transcript expression data in K562 cell line as transcripts per million (TPM) from the CCLE database of the transcripts of a target gene targeted by a gRNA were summed and the gRNAs with the highest sum and guide score were selected. For every step of the gRNA selection criteria, a minimum distance of 150 base pairs between target sites on the gene mRNA transcript was maintained to avoid targeting narrow regions of the mRNA transcripts. Finally, 30 non-target control gRNAs were selected from Wessels et al. [16] and added to the library (Supplementary Data 1). To

design the U6-a1-U6-a2 library, same gene targeting Cas13d arrays were then made from the 27 selected gRNAs by compiling 3 gRNAs from each group separated by Cas13d DR36 sequence to make 3 gRNAs arrays. Non-target control arrays were made by randomly compiling 3 non-target gRNAs to form 10 arrays using the same design. The U6-a1-U6-a2 library was therefore 9 fold smaller than the U6-g1-U6-g1 library (Supplementary Data 1).

The Cas9 sgRNA library consisted of 4 sgRNAs per gene with 10 non-target control sgRNA and 10 safe-cutters sgRNAs. The sgRNA sequences were selected from the Brunello genome-wide library (Supplementary Data 1)[45].

## Cas13d libraries cloning

The U6-g1-g2 library was cloned into the 783-Rx-mU6 plasmid. The selected 27 gRNAs per gene and 30 NTC gRNA were cloned so that only gRNA-gNTC, gNTC-gRNA, and gNTC-gNTC combinations are obtained. For that, gRNA template sequences for positions 1 and 2 of the format 5′-AGTATGCACAGTTGATCCGTCTCATTGG-DR36-$(N)_{23}$-CAAGTAAAC CCCTACCAACTAGAGACGGTTCACTGTAGCA-3′ and 5′-AGTATGCAC AGTTGATCCGTCTCAAACTGGtCGGGGTTTGAAAC-$(N)_{23}$-DR36-GCTT TAAAGAGACGGTTCACTGTAGCA, respectively, were designed. The final library consisted of 11,520 elements. For the U6-g1-U6-g2 and U6-a1-U6-a2 libraries were cloned into the 783-Rx-Dual plasmid. To systematically determine the effect of the different promoters used for gRNA/array expression on their activity, the libraries were cloned symmetrically such that the selected 30 gRNAs per gene and 30 NTC gRNA were cloned into both gRNA positions. gRNA and array template sequences of the format

5′-AGTATGCACAGTTGATCCGTCTCAAAAC-$(N)_{23}$-CAAGAGAGAC GGTTCACTGTAGCA-3′ and 5′-AGTATGCACAGTTGATCCGTCTCAAAA C-$(N)_{23}$-DR36-$(N)_{23}$-DR36-$(N)_{23}$-CAAGAGAGACGGTTCACTGTAGCA-3′ respectively were designed. The U6-g1-U6-g2 and U6-a1-U6-a2 final libraries consisted of 36,864 and 4096 elements respectively.

The oligo pools were synthesized as oligo pools (Twist Bioscience) and PCR-amplified using Phusion Flash High-Fidelity PCR Master Mix (ThermoFisher Scientific, F548L) according to the manufacturer's protocol with 0.1 ng/µL sgRNA template DNA, 1 µM forward primer (5′-GCCGTCTAATGTTCAGCTAGTATGCACAGTTGATCCGTCTC-3′), 1 µM reverse primer (5′- GTGTGACGTATGATCAGATCTATGCTACAGTGA ACCGTCTC-3′) in 50 µL total volume and the following cycle numbers: 1× (98 °C for 3 min), 16× (98 °C for 1 s, 64 °C for 15 s, 72 °C for 20 s) and 1× (72 °C for 5 min). PCR products were purified using NucleoSpin columns (Macherey-Nagel, 740609.250) followed by restriction digestion with BsmBI-v2 (NEB, R0739S) at 55 °C overnight. The digested fragments from the U6-g1-g1 and U6-a1-U6-a2 libraries were then run on a 2% agarose gel followed by excision of the digested band and purification via NucleoSpin columns (Macherey-Nagel, 740609.250). The U6-g1-U6-g2 library on the other hand had a fragment size of only 27 bp so it was run on a 20% Gradient TBE gel (Thermo Fischer Scientific, EC6315BOX). At the end of the run the gel was stained with SYBR-Gold (Thermo Fischer Scientific, S11494) and the digested fragment was cut. The gel pieces were passed through a microcentrifuge tube pierced with an 18-G needle. The gel slurry was resuspended in 400 µL water and incubated at 70 °C for 45 min. The gel was then removed by passing the mixture through a SpinX column (Sigma-Aldrich, CLS8162) after centrifugation at 20,000g for 3 min. The DNA fragments were then extracted by ethanol precipitation.

In parallel, the vectors mentioned above were prepared by restriction digestion with AarI (Thermo Fisher Scientific, ER1582) at 37 °C overnight. The digestion reaction was run on a 1% agarose gel followed by excision of the digested band and purification via NucleoSpin columns (Macherey-Nagel, 740609.250). 500 ng digested vectors and the amplified sgRNA library inserts were ligated at a 2:1 insert:vector ratio using T4 ligation in a 20 µL reaction at 16 °C overnight. The reaction was purified ethanol

precipitation and the resuspended volume was transformed into MegaX DH10β (Thermo Fisher Scientific, C640003) by electroporation using 100 ng of precipitated ligated DNA per 20 μL of bacterial suspension. *Escherichia coli* were recovered and cultured overnight in 100 mL LB (100 μg/mL ampicillin). The plasmid library was extracted using NucleoBond Xtra Midi kit (Macherey-Nagel, 740410.5). In parallel, a fraction of the transformation reaction was plated and used to determine the total number of transformed clones to ensure the coverage remained above 1000x. For the U6-g1-U6-g2 and the U6-a1-U6-a2 libraries, the cloned plasmids were then digested again with BfuA1 (NEB, R0701S) at 37 °C overnight and the library oligo fragments were cloned in position 2 by T4 ligation as before. The libraries U6-g1-g2, U6-g1-U6-g2, and U6-a1-U6-a2 had a final coverage of 2700x, 2700x, and 11,700x per library element respectively ensuring even representation of all library sequences and their narrow distribution.

### Cas9 library cloning

The selected 20-nt target specific sgRNA sequences were cloned into the pMB1 library vector by Gibson Assembly[44]. sgRNA template sequences of the format: 5′-GGAGAACCACCTTGTTGG-(N)20-GTTTAAGAGCTAAGCTGGAAAC-3′ were synthesized as oligo pools (Integrated DNA Technologies). The oligo pools were PCR-amplified using Phusion Flash High-Fidelity PCR Master Mix (ThermoFisher Scientific, F548L) according to the manufacturers protocol with 1 ng/μL sgRNA template DNA, 1 μM forward primer (5′-GGA-GAACCACCTTGTTGG-3′), 1 μM reverse primer (5′- GTTTCCAGCT-TAGCTCTTAAAC-3′) in 50 μL total volume and the following cycle numbers: 1× (98 °C for 3 min), 16× (98 °C for 1 s, 54 °C for 15 s, 72 °C for 20 s) and 1× (72 °C for 5 min). PCR products were purified using NucleoSpin columns (Macherey-Nagel, 740609.250). The library vector pMB1 was prepared by restriction digestion with AarI (Thermo Fisher Scientific, ER1582) at 37 °C overnight. The digestion reaction was run on a 1% agarose gel followed by excision of the digested band and purification via NucleoSpin columns (Macherey-Nagel, 740609.250). 100 ng digested pMB1 and 2.4 ng amplified sgRNA library insert was assembled using Gibson Assembly Master Mix (NEB, E2611L) in a 20 μL reaction for 30 min. The reaction was purified using P-30 buffer exchange columns (Bio-Rad, 7326250) that were equilibrated 5× with H2O and the eluted volume was transformed into 20 μL of MegaX DH10β (Thermo Fisher Scientific, C640003) by electroporation. *Escherichia coli* were recovered and cultured overnight in 100 mL LB (100 μg/mL ampicillin). The plasmid library was extracted using NucleoBond Xtra Midi kit (Macherey-Nagel, 740410.5). In parallel, a fraction of the transformation reaction was plated and used to determine the total number of transformed clones. The coverage was determined to be 3,111x clones per sgRNA ensuring even representation of all library sgRNA sequences and their narrow distribution.

### Pooled proliferation CRISPR screen

The K562-CasRx cells were transduced with lentivirally packaged sgRNA libraries at an MOI of >0.3 and a 1000-fold coverage. The low MOI was used to reduce the frequency of multiple infected cells, thus introducing one copy of the gRNA expression cassette per cell. The cells were then cultured in RPMI with 10% FBS and 1x Pen/Strep (Sigma-Aldrich, P0781-100ML) in a 37 °C incubator with 5% CO2. 24 h post-transduction, the cells were selected with Puromycin (Carl Roth, 0240.2) (2 μg/mL) for 96 h. After selection, aliquots of cells representing 1000x coverage each were centrifuged at 1000*g* for 5 min, and the pellets were frozen for later analysis using NGS (see below). The cell numbers representing 1000x coverage for the different screens were as follows: U6-g1-g2 with 11,520,000 cells, U6-g1-U6-g2 with 36,864,000 cells, U6-a1-U6-a2 with 4,096,000 cells, and the

Cas9 screen with 36,000 cells. The remaining cells were diluted to a density of 100,000 cells/mL with fresh medium. The cells were split into 2 fractions. One fraction was imatinib treated and the other fraction represented untreated cells. An IC50 concentration of 100 nM Imatinib (MedChemExpress, HY−50946) was added to the treated cell culture plates. Imatinib was renewed at day 3 of cell splitting (IC60 = 150 nM) and day 7 (IC80 = 300 nM) following the start of the imatinib treatment. Cells for endpoint analysis were harvested on day 19. On day 19, cells representing 1000x coverage of the libraries per sample were harvested for downstream analysis using NGS as described below. Coverage at the cell level was maintained over 1000x throughout the screens, and the culture was diluted with fresh medium when the cell density reached 1 million cells/mL.

### Genomic DNA extraction

For the U6-g1-g2 and the U6-g1-U6-g2 screens, the cell pellets from the baseline and day 19 time point untreated and imatinib-treated samples were resuspended in 20 ml of P1 buffer (Qiagen, 19051) containing 100 μg/ml RNase A (Sigma-Aldrich, 10109142001) and 0.5% SDS (Sigma-Aldrich, 71725-50 G), followed by incubation at 37 °C for 30 min. Then, Proteinase K (Sigma-Aldrich, 70663-4) was added (final concentration 100 μg/ml) and incubated at 55 °C for 30 min. After digestion, the samples were homogenized by passing through an 18 G needle three times and then through a 22 G needle three times. The homogenized samples were mixed with 20 ml Phenol:Chloroform:Isoamyl alcohol (Thermo Fisher Scientific, 15593031), transferred to 50 ml MaX-tract tubes (Qiagen, 129073), and thoroughly mixed. The samples were then centrifuged at room temperature (RT) for 5 min at 1500*g*. The aqueous phase was transferred to ultracentrifuge tubes and thoroughly mixed with 2 ml of 3 M sodium acetate (Sigma-Aldrich, S2889-250G) plus 16 ml of isopropanol (Sigma-Aldrich, 650447-1 L) at RT before centrifugation at 15,000*g* for 15 min. The gDNA pellets were gently washed with 10 ml of 70% ethanol (Thermo Fischer Scientific, 17740239) and dried at 37 °C. The dry pellets were resuspended in H2O, and the gDNA concentration was adjusted to 1 μg/μl. The degree of gDNA fragmentation was assessed on a 1% agarose gel, and the gDNA was further fragmented by boiling at 95 °C until the average size ranged between 10 and 20 kb.

For the U6-a1-U6-a2, and the Cas9 screens, the genomic DNA was extracted from the baseline and day 19 time point untreated and imatinib-treated samples using the DNeasy Blood & Tissue DNA Purification kit (Qiagen, 69506) according to the manufacturer's instructions.

### PCR recovery of gRNA/array sequences

Two nested PCR reactions were performed to amplify the U6-g1-g2, U6-g1-U6-g2, and the Cas9 screens gRNA/array cassette from the extracted gDNA. For the first PCR reactions, up to 50 μg gDNA, 0.3 μM forward (5′-GGCTTGGATTTCTATAACTTCGTATAGCA-3′) and reverse (5′-CGGGGACTGTGGGCGATGTG-3′) primer, 200 μM dNTP mix (Thermo Fischer Scientific, 10297018), 1x Titanium Taq buffer and 2 μL Titanium Taq polymerase (Takara, 639209) were mixed in 50 μL total volume. For the U6-g1-U6g1 screen samples, the PCR reaction was run using the following cycles: 1x (95 °C, 3 min), 16x (95 °C, 30 s, 62 °C, 30 s, 68 °C, 3 min), 1x (68 °C, 5 min). For the U6-g1-g2 and the Cas9 screens, the PCR reaction cycling conditions were 1x (94 °C, 3 min), 16x (94 °C, 30 s, 62 °C, 10 s, 72 °C, 20 s), 1x (68 °C, 2 min). For the second PCR reactions, 2 μL first-round PCR, 0.5 μM forward (5′-AATGA-TACGGCGACCACCGAGATCTACACACACTCTTTCCCTACACGACGCT CTTCCGATCTTGAGACTATAAGTATCCCTTGGAGAACCACCTTG-3′ and 5′-AATGATACGGCGACCACCGAGATCTACACACACTCTTTCCCTA CACGACGCTCTTCCGATCTTCCCTTGGAGAACCACCTTGTTGG-3′ for the Cas13d and Cas9 screens samples respectively) and reverse

(5′-CAAGCAGAAGACGGCATACGAGA-(N)$_6$-TGTGACTGGAGTTCAGA CGTGTGCTCTTCCGATCTATTGCTAGGACCGGCCTTAAAGC-3′ and 5′-CAAGCAGAAGACGGCATACGAGAT-(N)$_6$-GTGACTGGAGTTCAGACG TGTGCTCTTCCGATC-3′ for the Cas13d and Cas9 screens samples respectively) primer where (N)$_6$ is a 6 nt index for sequencing on the Illumina NGS platform, 200 μM dNTP mix (Thermo Fischer Scientific, 10297018), 1x Titanium Taq buffer and 1.5 μL Titanium Taq (Takara, 639209). For the U6-g1-U6g1 screen samples, the PCR cycles were: 1x (95 °C, 3 min), 12x (95 °C, 30 s, 55 °C, 30 s, 68 °C, 3 min), 1x (68 °C, 5 min). For the U6-g1-g2 and the Cas9 screens, the PCR reaction cycling conditions were 1x (94 °C, 3 min), 20x (94 °C, 30 s, 55 °C, 10 s, 72 °C, 20 s), 1x (68 °C, 2 min). The PCR products for the U6-g1-g2, U6-g1-U6-g2, and the Cas9 screens had a size of 344 bp, 888 bp, and 325 bp respectively and were purified from a 1% agarose gel via NucleoSpin columns (Macherey-Nagel, 740609.250).

For the U6-a1-U6-a2 screen samples, a 1-step PCR reaction was performed using ExTaq Polymerase (Takara, RR001A). For that, 10 μg gDNA, 0.5 μM forward (5′-AATGATACGGCGACCACCGAGATCTACA-CACACTCTTTCCCTACACGACGCTCTTCCGATCTTGAGACTATAAGT ATCCCTTGGAGAACCACCTTG-3′) and reverse (5′-CAAGCAGAAGACG GCATACGAGA-(N)$_6$-TGTGACTGGAGTTCAGACGTGTGCTCTTCCGAT CTATTGCTAGGACCGGCCTTAAAGC-3′) primer, 800 μM dNTP mix, 1x ExTaq buffer and 1.5 μL ExTaq (Takara) were mixed and the PCR cycles were set at 1x (95 °C, 1 min), 28x (95 °C, 30 s, 66 °C, 30 s, 72 °C, 1 min), 1x (72 °C, 10 min). The PCR product had a size of 1124 bp and was purified from a 1% agarose gel via NucleoSpin columns (Macherey-Nagel, 740609.250). NGS for all amplified samples was performed on either MiniSeq or NovaSeq 6000 Illumina platforms.

## Cas9 screen data analysis

Paired-end sequencing was performed using a NovaSeq 6000 platform. The data analysis was performed using MAGeCK 0.5.9.2[50]. In short, sgRNA read count files were computed from the raw CRISPR FASTQ files using the count function (Supplementary Data 4). The MAGeCK MLE command was then used to calculate the MAGeCK beta score, Wald-P values and false discovery rates for the enrichment and depletion of each guide were compared to the baseline sample.

## Cas13d screens data analysis

Paired-end sequencing was performed using a NovaSeq 6000 and NovaSeq X Plus for U6-g1-g2.

We started the analysis of the proliferation screens by using MAGeCK for gRNA counting. Initially, we filtered the FASTQ files from the U6-g1-U6-g2 and U6-a1-U6-a2 screen samples to exclude sequences containing the 5′-GCTTTAAGGC-3′ in read 1 and the 5′-CCAACAAGGT-3′ in read 2 to remove recombined constructs expressing a single gRNA (sgRNA) or single-gene array (array). Following this, cutadapt 4.4[51] was used to remove regions spanning the first 74 bp and the last 18 bp in read 1, and the first 59 bp and the last 33 bp in read 2, isolating gRNA-1 and gRNA-2 in U6-g1-g2 and U6-g1-U6-g2, respectively, and gRNA-1 from array-1 and gRNA-3 from array-2 in U6-a1-U6-a2. The resulting trimmed reads were merged using FLASH v1.2.11[52]. In order to accommodate up to three mismatches for read mapping we used Bowtie 1.2.3[53]. Conversion of resulting sam files to bam files was performed using SAMtools 1.18[54], which served as input for MAGeCK COUNT (Supplementary Data 4). The distribution of the libraries was then determined from the baseline samples.

Based on the count matrix, we then calculated the tau values and GI scores (Supplementary Data 2). To eliminate low read counts we systematically removed gRNA combinations within each screen that had counts below 10% of the baseline mean counts. Additionally, to mitigate the occurrence of zero counts, we applied a pseudocount to each count value.

Normalization was performed by dividing each count by the mean value of the corresponding sample. Tau values were then computed

using the following formula:

$$\tau_x = log_2 \left( \frac{\left( \frac{N_t^x}{N_{t_0}^x} \right)}{\left( \frac{N_t^{NTC}}{N_{t_0}^{NTC}} \right)} \right) \qquad (1)$$

where $N^x$ denotes the frequency of sgRNA $x$ and $N^{NTC}$ denotes the frequency of non-targeting control gRNAs before ($t_0$) or after ($t$) imatinib treatment.

To consider the non-zero gNTC*gNTC tau values within the tau values of the sgRNA and array combinations, we determined the average tau values of the gNTC*gNTC combinations for each sample and subtracted this average from all tau values within that specific sample.

Afterwards, we calculated the GI score for each combination using the following formula:

$$GI = \tau_{(Gene1+Gene2)} - \left( \tau_{Gene1} + \tau_{Gene2} \right) \qquad (2)$$

where $\tau_{Gene1+Gene2}$ is the measured phenotype of the double perturbation and $\tau_{Gene1} + \tau_{Gene2}$ is the expected phenotype calculated from the measured individual perturbation phenotype of each gene.

To obtain stable GI scores, we removed sgRNAs and arrays with weak phenotypes. Therefore, we first calculated the mean value of tau scores per sgRNA/array combination across both replicates of imatinib treatment. Then, for each gene per sgRNA/array, the mean across all sgRNA-NTC/array-NTC phenotypes was calculated. A distinction was made between the respective orientation of sgRNA/array in position 1 (sgRNA-NTC/array-NTC) or sgRNA/array in position 2 (NTC-sgRNA/ NTC-array). The individually calculated mean values for each sgRNA/ array were then used to determine the overall mean value per gene, which serves as a cut-off value for the determination of weaker phenotypes. For each gene and orientation, all sgRNAs/arrays that did not fulfill the cut-off value were determined. Finally, GI scores of sgRNAs/ arrays that did not meet the cut-off value for both orientations were removed.

Analogous to Aregger et al.[55] all remaining GI scores per gene were mean-summarized and their significance was calculated using limma moderated $t$-test followed by Benjamini-Hochberg multiple testing correction (Supplementary Data 5)[55].

The calculation of the "same gene GI" and the "single gene controls" was performed in the same way as the calculation for the GI scores described above. "Same gene GIs" represent "self-genetic interactions" of a gene such as ABL1-ABL1. For a better understanding, the formula for the calculation is shown below using ABL1 as an example:

$$GI = \tau_{(ABL1+ABL1)} - \left( \tau_{ABL1} + \tau_{ABL1} \right) \qquad (3)$$

In contrast, "single gene controls" show the interaction of a gene with all non-target controls, such as ABL-NTC. While we do not consider orientation in "same gene GI", it is considered in "single gene controls".

$$GI = \tau_{(ABL1+NTC)} - \left( \tau_{ABL1} + \tau_{NTC} \right) \qquad (4)$$

All scripts used for data analysis were written in R 4.3.0[56]. To create the plots shown in the figures we used the R package ggplot2[57] and ComplexHeatmap[58] as well as GraphPad Prism 10.3.0. The network was created with BioRender.

## Statistics and reproducibility

For all experiments, the number of technical and/or biological replicates is listed in the figure legends or text. Unless otherwise indicated,

statistical significance was calculated using limma moderated *t* test followed by Benjamini-Hochberg multiple testing correction. Pearson correlation was used to determine the *r* values. Statistical analyses were performed using GraphPad Prism 9 (GraphPad Software) or the R language programming environment.

## Reporting summary

Further information on research design is available in the Nature Portfolio Reporting Summary linked to this article.

## Data availability

Raw FASTQ files have been deposited on the SRA database under the following: accession code PRJNA1092399. Plasmids and their sequences are deposited at Addgene: pXR001-mCD4 (Addgene #228359), 783-Rx-hU6 (Addgene #228360), 783-Rx-mU6 (Addgene #228361), 783-Rx-Dual (Addgene #228362), pMB1 (Addgene #228363), AiO-Cas12a (Addgene#228364). Source data are provided with this paper.

## Code availability

Data processing scripts and raw input data for the data processing scripts are available at Zenodo [https://doi.org/10.5281/zenodo.13841429]. The code is released under the MIT License [https://opensource.org/licenses/MIT].

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

## Acknowledgements

This work was supported by the European Union through the ESF grant ZS/2016/08/80642.

## Author contributions

G.E.K. performed all experiments. J.H. performed data analysis. M.B. conceived the study, designed experiments, supervised the work and acquired funding. G.E.K., J.H., M.B. wrote and revised the manuscript.

## Funding

## Competing interests

The authors declare no competing interests.
