## [Transparent Peer Review file · Nature Communications]

Evaluation of Cas13d as a tool for genetic interaction mapping

Corresponding Author: Professor Michael Boettcher

This manuscript has been previously reviewed at another journal. This document only contains information relating to versions considered at Nature Communications. Mentions of prior referee reports have been redacted.

Version 1:

Reviewer comments:

Reviewer #1

(Remarks to the Author)

I support the publication of this transferred and previously reviewed manuscript at Nature Communications.

(Remarks on code availability)

NA

Reviewer #2

(Remarks to the Author)

Summary

Overall, I really appreciate the work that went into the revision of the manuscript and I now think it much better conveys this very timely subject. Most of my comments have been addressed. However, I would like to deepen the discussion of one major concern that, I think, can be addressed.

Reviewer #2: Major remarks

Comment 1: My concerns are addressed

Comment 2: **Redacted**

(Current comment) I appreciate that the authors have clarified the presentation of data reproducibility. I would like to be clear that none of the statements in this manuscript are relevant without understanding to which level they generalize, i.e., reproduce. Therefore, I would like to stress that it is important to display reproducibility at different levels to judge the quality of the data well and recommend to emphasize the reproducibility of genetic interactions and, where suitable, tau scores in the main figures since this is the data processing level of interest. At the same time, it would be advisable to also show reproducibility of all other data processing levels, such as read counts, remaining tau scores, etc., in a less prominent place (e.g., the supplements).

Comment 3&6: My concerns are addressed.

Comment 4: My concerns are addressed.

Comment 5: My concerns are addressed.

Minor remarks

Comment 1: My concerns are addressed.

Comment 2: My concerns are addressed.

Comment 3: My concerns are addressed.

Comment 4: My concerns are addressed.

Comment 5: My concerns are addressed.

Comment 6: My concerns are addressed.

Comment 7: **Redacted**

(Current comment) I agree that adding a significance measure such as an FDR to the genetic interactions in Figure 3D and this would address my concerns. However, unless the reproducibility is not explicitly measured here, it should not be described as reproducible genetic interactions. The term significant is more appropriate.

Comment 8: My concerns are addressed.

(Remarks on code availability)

I have only skimmed over the code structure and elements made available. The code is a useful resource for reproducing the presented work.

Reviewer #3

(Remarks to the Author)

Kassem et al. have responded to some but not all my concerns. My main concern remains that the utility of this *Ruminococcus flavefaciens* Cas13d genetic interaction mapping approach requires that A) Cas13d perturbations be highly specific in the context of single gRNAs and also single gene arrays and B) that there be a robust understanding of how collateral cleavage toxicity impacts GI measurements. As stated previously, the discoverers of Cas13d (as well as multiple other groups) have documented the off-target activity and collateral cleavage toxicity of Cas13d (CasRx) even when only targeting 1 gene with one gRNA (see for example figures 4 and 5 in PMID: 38091991). These problems will be compounded by targeting 2 genes with 2 or more gRNAs. The authors need to directly address these concerns by cell growth/cell cycle/cell death measurements and also RNA seq. Comparison to Cas9 phenotypes is insufficient and indirect. They claim Cas13d has no effect on cell death but don't show any data and they don't have controlled experiments that examine cell health/growth or targeting specificity.

The comparisons relevant for validating that their GI approach does not have collateral toxicity and is highly specific in transcript targeting need to be made against both NTC controls and also against empty vector (no gRNAs). Or the comparisons could be made +/- Cas13d. It is now well established that collateral cleavage toxicity associated with Cas13d relates to the abundance of the target transcript (see PMIDs: 35244715, 36977923, 38091991) and so toxicity must be addressed against genes with a range of expression including most importantly for genes with high expression (see as an example PMID: 38091991 Figure 4F&G). The reason this is so critical for this manuscript is because in a GI experiment the collateral cleavage toxicity could be variable across different gene pairs based on the level of expression of each pair of genes.

As a less critical but still important point, the authors need to address that the literature shows that different cell types can have dramatically different sensitivities to collateral cleavage (PMID:38091991) as it is not helpful or useful to publish a genetic interaction mapping approach that is solely useful in K562 cells.

(Remarks on code availability)

Version 2:

Reviewer comments:

Reviewer #2

(Remarks to the Author)

My comments have now been addressed in full and I support the publication of this work at Nature Communications.

(Remarks on code availability)

Reviewer #3

(Remarks to the Author)

the authors have addressed my concerns by modifying the text to A) highlight their data which shows that in this specific context collateral cleavage toxicity is not a major concern and B) add relevant text and references which highlight the concerns around toxicity and also to point to orthologs or other engineered versions of Cas13d that might get around this issue.

(Remarks on code availability)

On the utility of Cas13d for genetic interaction mapping

Ghanem El Kassem¹, Jasmine Hillmer¹, Michael Boettcher¹

¹ Universitätsmedizin Halle, Martin Luther University Halle-Wittenberg, Halle (Saale), 06120, Halle, Germany

We would like to thank the reviewers once again for their feedback on our manuscript. In the following, we address each comment point by point.

Reviewer #2:

Summary

Overall, I really appreciate the work that went into the revision of the manuscript and I now think it much better conveys this very timely subject. Most of my comments have been addressed. However, I would like to deepen the discussion of one major concern that, I think, can be addressed.

We thank the reviewer for appreciating the effort we have put into revising the manuscript and agree that the key findings are now much better conveyed.

Reviewer #2: Major remarks

Comment 2:

Redacted

(Current comment) I appreciate that the authors have clarified the presentation of data reproducibility. I would like to be clear that none of the statements in this manuscript are relevant without understanding to which level they generalize, i.e., reproduce. Therefore, I would like to stress that it is important to display reproducibility at different levels to judge the quality of the data well and recommend to emphasize the reproducibility of genetic interactions and, where suitable, tau scores in the main figures since this is the data processing level of interest. At the same time, it would be advisable to also show reproducibility of all other data processing levels, such as read counts, remaining tau scores, etc., in a less prominent place (e.g., the supplements).

We agree with the reviewer's comment and have now added reproducibility data at all possible levels (normalized read counts, tau and GI scores for both untreated and imatinib treated screens). In detail, we have added:

Supp. Fig. 1 shows the reproducibility of the U6-g1-g2 screens. (A-B) Correlation between normalized read counts from two technical screen replicates in the untreated (A) and imatinib treated (B) conditions. (C) Correlation between tau values from two technical screen replicates in the untreated condition before filtering for functional gRNAs. (D-E) Correlation between tau values from two technical screen replicates in the untreated (D) and imatinib treated (E) condition after filtering for functional gRNAs. (F) Correlation between tau values from gRNA-gNTC and gNTC-gRNA combinations in the untreated condition. Correlation plot between tau values from two technical screen replicates before filtering for functional gRNAs and the correlation plot between tau values from gRNA-gNTC and gNTC-gRNA combinations following 19 days of imatinib treatment are shown in Fig. 2C and 2D.

Supp. Fig. 3 shows the reproducibility of the U6-g1-U6-g2 screens. (A-B) Correlation between normalized read counts from two technical screen replicates in the untreated (A) and imatinib treated (B) conditions. (C) Correlation between tau values from two technical screen replicates in the untreated condition before filtering for functional gRNAs. (D-E) Correlation between tau values from two technical screen replicates in the untreated (D) and imatinib treated (E) condition after filtering for functional gRNAs. (F) Correlation between tau values from gRNA-gNTC and gNTC-gRNA combinations in the untreated condition. Correlation plot between tau values from two technical screen replicates before filtering for functional gRNAs and the correlation plot between tau values from gRNA-gNTC and gNTC-gRNA combinations following 19 days of imatinib treatment are shown in Fig. 2G and 2H.

Supp. Fig. 5 shows the reproducibility of the U6-a1-U6-a2 screens. (A-B) Correlation between normalized read counts from two technical screen replicates in the untreated (A) and imatinib treated (B) conditions. (C-D) Correlation between tau values from two technical screen replicates in the untreated (C) and imatinib treated (D) condition before filtering for functional gRNAs. (E-F) Correlation between tau values from two technical screen replicates in the untreated (E) and imatinib treated (F) condition after filtering for functional arrays. (G-H) Correlation between tau values from gRNA-gNTC and gNTC-gRNA combinations in the untreated (G) and imatinib treated (H) condition.

Supp. Fig. 6 (previously Supp. Fig. 4) shows the reproducibility of the GI scores calculated from the tau values of U6-g1-U6-g2 and U6-a1-U6-a2 screens. (A-B) Pearson correlation between GI scores determined from (A) single gRNA and (B) single-gene array screens, without (top) and with imatinib treatment (bottom). The left panels show the correlation of GI scores between Gene1-Gene2 and Gene2-Gene1 orientations. The right panels show the correlation of GI scores between screen replicates.

Comment 7:

Redacted

(Current comment) I agree that adding a significance measure such as an FDR to the genetic interactions in Figure 3D and this would address my concerns. However, unless the reproducibility is not explicitly measured here, it should not be described as reproducible genetic interactions. The term significant is more appropriate.

By reproducibility, we actually meant to refer to GIs that came out as significant in both gRNA orientations (A-B and B-A), meaning that they could be “reproduced” in either orientation. This has now been made more clear in the text by adding the sentence: “Genetic interaction network of reproducible GIs in both gene orientations (Gene1-Gene2 and Gene2-Gene1) that were identified between all six investigated genes derived from the single-gene array strategy. Only interactions that were identified by GIs of $> \pm 0.2$ and FDR < 0.5 in both orientations (gRNA-gNTC and gNTC-gRNA) are shown.”

Reviewer #2 (Remarks on code availability):

I have only skimmed over the code structure and elements made available. The code is a useful resource for reproducing the presented work.

We appreciate the acknowledgment of the usefulness of the submitted code by the reviewer.

Reviewer #3 (Remarks to the Author):

Kassem et al. have responded to some but not all my concerns. My main concern remains that the utility of this *Ruminococcus flavefaciens* Cas13d genetic interaction mapping approach requires that A) Cas13d perturbations be highly specific in the context of single gRNAs and also single gene arrays and B) that there be a robust understanding of how collateral cleavage toxicity impacts GI measurements. As stated previously, the discoverers of Cas13d (as well as multiple other groups) have documented the off-target activity and collateral cleavage toxicity of Cas13d (CasRx) even when only targeting 1 gene with one gRNA (see for example figures 4 and 5 in PMID: 38091991). These problems will be compounded by targeting 2 genes with 2 or more gRNAs. The authors need to directly address these concerns by cell growth/cell cycle/cell death measurements and also RNA seq. Comparison to Cas9 phenotypes is insufficient and indirect. They claim Cas13d has no effect on cell death but don't show any data and they don't have controlled experiments that examine cell health/growth or targeting specificity.

The comparisons relevant for validating that their GI approach does not have collateral toxicity and is highly specific in transcript targeting need to be made against both NTC controls and also against empty vector (no gRNAs). Or the comparisons could be made +/- Cas13d. It is now well established that collateral cleavage toxicity associated with Cas13d relates to the abundance of the target transcript (see PMIDs: 35244715, 36977923, 38091991) and so toxicity must be addressed against genes with a range of expression including most importantly for genes with high expression (see as an example PMID: 38091991 Figure 4F&G). The reason this is so critical for this manuscript is because in a GI experiment the collateral cleavage toxicity could be variable across different gene pairs based on the level of expression of each pair of genes.

We are pleased that we were able to address all but one of reviewer no. 3's concerns. We agree that the reports of collateral RNA cleavage activity with *Ruminococcus flavefaciens* Cas13d (RfxCas13d) are important to consider. In our revised Discussion section, we have therefore further acknowledged the potential for collateral activity, now specifically citing all relevant studies noted by the reviewer (e.g., PMIDs: 35244715, 36977923, 38091991).

We respectfully disagree though, with the reviewer's assertion that we lack data to rule out collateral RNA cleavage in our study. As previously shown in the paragraph "*Cas13d perturbation shows no non-specific proliferation phenotypes*" and Supplementary Figure 2 (now Supplementary Figure 4), we present proliferation data from four pooled Cas13d screens targeting the six genes used throughout this study. For the non-essential genes PTPN1, NF1, and SPRED2, we observe no proliferation phenotype when compared to non-target control gRNAs in the untreated arm of the screen, both when targeted with single gRNAs (Fig. S4A) and with gRNA arrays (Fig. S4B). These findings align with the Cas9 data from our screens and the DepMap, providing evidence against cytotoxicity from Cas13d collateral activity. Additionally, we observe the expected enrichment of PTPN1-, NF1-, and SPRED2-perturbed cells under imatinib treatment, demonstrating that effective perturbation of these genes with single gRNAs (Fig. S4C) and gRNA arrays (Fig. S4D) produces the anticipated imatinib resistance phenotypes. Furthermore, if collateral RNA cleavage was occurring in our system, targeting a transcript would cause nonspecific trans-cleavage of other transcripts and their corresponding proteins. However, the flow cytometry data shown

in Figure 1D, demonstrates that targeting CD46 or CD47 individually has no impact on the expression levels of the respective other gene.

Together these findings argue strongly against collateral RNA cleavage being an issue in our study, which is consistent with numerous studies that have utilized RfxCas13d in many different cell types (e.g., PMIDs: 38409225, 32518401, 36550277, 35044815, 39302832, 39288267, 38609360), including the chronic myeloid leukemia cell line K562, used here (Liang et al., Cell 2024), without adverse effects from collateral activity. Additionally, a recent study from Stanley Qi's lab demonstrated the simultaneous perturbation of up to 10 different genes in primary human T cells, with no signs of collateral activity. This suggests that RfxCas13d can achieve highly multiplexed gene perturbation without inducing collateral effects (PMID: 38387457).

We now also highlight in the Discussion the availability of high-fidelity RfxCas13d variants with reduced collateral activity (PMID: 35953673) as well as Cas13d orthologs from different bacterial species, such as DjCas13d (PMID: 38091991), which offer promising options for future studies in cell types that may be more sensitive to collateral activity.

As a less critical but still important point, the authors need to address that the literature shows that different cell types can have dramatically different sensitivities to collateral cleavage (PMID:38091991) as it is not helpful or useful to publish a genetic interaction mapping approach that is solely useful in K562 cells.

This point is addressed now in a paragraph about collateral activity in the revised Discussion.